# Cannabidiol on the Path from the Lab to the Cancer Patient: Opportunities and Challenges

**DOI:** 10.3390/ph15030366

**Published:** 2022-03-17

**Authors:** Miguel Olivas-Aguirre, Liliana Torres-López, Kathya Villatoro-Gómez, Sonia Mayra Perez-Tapia, Igor Pottosin, Oxana Dobrovinskaya

**Affiliations:** 1Laboratory of Immunobiology and Ionic Transport Regulation, Centro Universitario de Investigaciones Biomédicas, Universidad de Colima, Colima 28045, Mexico; miguel.a.olivas@gmail.com (M.O.-A.); torres_liliana@ucol.mx (L.T.-L.); kathya8899@hotmail.com (K.V.-G.); 2Unidad de Desarrollo e Investigación en Bioterapeúticos (UDIBI), Escuela Nacional de Ciencias Biológicas, Instituto Politécnico Nacional (ENCB-IPN), Mexico City 11340, Mexico; mayra.perez@udibi.com.mx

**Keywords:** anticancer activity, anticancer treatment, adverse effects, cannabidiol, clinical trials, drug delivery, palliative care, pre-clinical studies, synergism

## Abstract

Cannabidiol (CBD), a major non-psychotropic component of cannabis, is receiving growing attention as a potential anticancer agent. CBD suppresses the development of cancer in both in vitro (cancer cell culture) and in vivo (xenografts in immunodeficient mice) models. For critical evaluation of the advances of CBD on its path from laboratory research to practical application, in this review, we wish to call the attention of scientists and clinicians to the following issues: (a) the biological effects of CBD in cancer and healthy cells; (b) the anticancer effects of CBD in animal models and clinical case reports; (c) CBD’s interaction with conventional anticancer drugs; (d) CBD’s potential in palliative care for cancer patients; (e) CBD’s tolerability and reported side effects; (f) CBD delivery for anticancer treatment.

## 1. Introduction

Cannabidiol (CBD) is the most abundant natural cannabinoid found in cannabis plants. The advantage of CBD is the apparent lack of any intoxicating effect. CBD has been proposed for the treatment of pain, insomnia, several psychological conditions, graft-versus-host disease, inflammatory diseases, and cancer [1,2,3,4,5,6,7]. The wide spectrum of biological effects seems to be related to numerous molecular targets for CBD, which include various G-protein-coupled receptors, ion channels and ionotropic receptors, transporter proteins, nuclear receptors, and numerous enzymes involved in lipid, xenobiotic/drug, and mitochondrial metabolism [7,8]. The anticancer properties of CBD are mostly reported in studies in vitro, and to a lesser extent in vivo, whereas clinical studies including cancer patients are still scarce. The goal of the present review is a critical assessment of CBD’s potential for anticancer therapy, recent advances, and challenges.

## 2. CBD Shows Anticancer Properties in Pre-Clinical Studies In Vitro and In Vivo

In the last several years, there has been growing interest in the use of cannabinoids in the treatment of various types of cancer. Two of them, CBD and Δ-9-tetrahidrocannabinol (THC), have demonstrated pronounced anticancer activity in pre-clinical in in vitro and in vivo trials. Because the use of THC in chemotherapy is limited due to its psychotropic effects, special attention is paid to the non-psychoactive CBD, which also has demonstrated a greater antitumor effect than THC [9,10,11]. A recent comprehensive review summarizes the biological effects of CBD in different tumor types and is highly recommendable for interested readers [6]. The biological effects of CBD have been tested in a broad range of tumor cells in vitro and in vivo (Table A1 in Appendix A), including glioma/glioblastoma [9,11,12,13,14,15,16,17,18,19], breast cancer [9,20,21,22,23,24,25,26], prostate carcinoma [9,27,28,29], leukemia/lymphoma cells [9,10,25,30,31,32,33], gastric cancer [9,34,35], colon/colorectal cancer [9,36,37,38], lung cancer [39,40,41], cervical cancer [25,39,42]), neuroblastoma [43,44], medulloblastoma [45], ependymoma [45], pancreatic cancer [46,47], ovarian cancer [28], endometrial cancer [48], bladder urothelial carcinoma [49], and head and neck squamous cell carcinoma [50].

Anticancer drug candidates are initially screened on cell lines, to reveal their biological effects and underlying molecular mechanisms. The antitumoral activity of CBD was tested in vitro in a wide range of concentrations, from 0.01 to 100 µM (Table A1). However, the variation in culture conditions (passage number, medium composition, presence of serum and cellular confluence) and the mode of CBD administration (single or repetitive daily administration) hinder direct data comparison and assessment of the relative sensitivity of cell lines to CBD. Lymphoblastic leukemia, particularly of T lineage, presents higher sensitivity to CBD when compared to myeloid leukemia and breast and cervical cancer, as was demonstrated in comparative viability assays carried out under the same experimental conditions [25]. To understand the antitumor action of CBD, it is necessary to consider the sequence of events induced in target cells and their interrelation. Table A1 summarizes cellular targets and processes in different cancer models affected by CBD at the time range from minutes to days, whereas a synthetic timeline for CBD’s biological effects, related to its antitumor activity, is given in Figure 1.

There are only a few studies that monitored the earliest/instantaneous responses to CBD. A rapid rise in the cytosolic free calcium (Ca^2+^) level ([Ca^2+^]_i_), which occurs within the first 3–5 min after CBD administration, was observed in breast cancer [9] and leukemic T cells [25]. In the latter work, concurrent measurements of mitochondrial Ca^2+^ ([Ca^2+^]_m_) and [Ca^2+^]_i_ revealed that the [Ca^2+^]_I_ rise was preceded by [Ca^2+^]_m_ transience, indicating the early involvement of mitochondria in the process [25]. Accordingly, the rapid dissipation of mitochondrial membrane potential (ΔΨm) and cytochrome C (Cyt C) release from mitochondria to cytosol was observed in this model during the first 10–20 min [25].

At physiological conditions, mitochondria are major contributors to reactive oxygen species (ROS) production. They also possess an efficient antioxidant enzyme system for rapid ROS scavenging, to prevent cell damage. Accordingly, mitochondrial disturbances are related to the increased production of ROS and oxidative stress [51]. Augmented ROS levels were reported from the first hour of CBD administration in different models, such as murine thymoma [33], human breast cancer [9], and T cell leukemia [25], and could be detected also at longer (24–96 h) times of observation (Table A1). Increased ROS production seems to be an important mediator of CBD cytotoxicity. ROS scavengers α-tocopherol (αTOC) and *N*-acetylcysteine (NAC) counteracted the antiproliferative effects of CBD in human glioblastoma [11,12], breast cancer [9,21], T cell leukemia [31], and mouse medulloblastoma [45]. Accordingly, αTOC rescued cancer cells from apoptosis [15,18,22,31].

Since mitochondrial damage causes reduced ATP production, the energy-consuming basic cellular functions such as migration are affected by CBD. Decreased migration capacities were reported from the first hour of CBD treatment in glioblastoma [13] and leukemia [25] and during 24–48 h of observation in multiple cancer types such as bladder carcinoma [49], neuro- and glioblastoma [17,41,43], breast [20,21], cervical [39], lung [39], and endometrial cancer [48], and squamous cell carcinoma [50].

Among early events, which developed within minutes after CBD administration, decreased levels of active (phosphorylated) AKT were reported [44], which in turn can be related to the inhibition of cellular metabolism and proliferation. Inhibition of AKT/mTOR and upregulation of MAPK signaling pathways were confirmed in many models (Table A1, Figure 1). Since decreased p-AKT levels are a key signal for the activation of autophagy [52] and are related to the upregulation of MAPK p38 [18], p-AKT downregulation is maintained over time [16,18,19,22,32] and correlates with the rise in p-p38 [18,19,39], decrement in p-mTOR [32], upregulation of key autophagic genes [16], and induction of autophagy [25,44]. Autophagy, depending on its scale, moderate or large, may act as a protective mechanism or can eventually lead to cell death [52].

In many cell types, CBD predominantly evoked apoptosis, as was evidenced by an increase in the expression/function of pro-apoptotic initiators (Bad, tBid) and pore-forming proteins (BAX), a decrease in antiapoptotic Bcl-2 [16,22], Cyt C release [14,22,25,31,34], and the activation of caspases [9,14,22,25,31,35,37]. Of note, in many studies, the type of cell death was not specified because only metabolic assays were performed. Thus, apoptosis likely is not a unique process induced by CBD but can be paralleled and/or affected by concurrent processes such as autophagy and metabolic inhibition. For example, although apoptosis was triggered first by CBD in leukemic cells, severe mitochondria damage and oxidative stress caused the switch to the mitochondrial permeability transition pore (mPTP)-driven necrosis [25].

There is plenty of evidence that CBD can strike multiple cellular targets. CBD possesses low affinity for classical cannabinoid receptors CB1 and CB2 but can efficiently antagonize their agonists; it also acts as a CB2 inverse agonist. Meanwhile, many of the CBD-mediated cellular effects are independent of the endocannabinoid system receptors. CBD acts as an antagonist of G-protein-coupled receptor GPR55, and as an agonist for serotonin receptor 5HT and transient receptor potential vanilloid receptors/channels TRPV1 and TRPV2 [2,6,7,53]. In addition, CBD is a small and lipophilic molecule. Thus, it easily permeates the plasma membrane, being able to reach intracellular targets as well. Accordingly, the mitochondrial outer membrane voltage-dependent anion channel (VDAC) was reported as a highly relevant CBD target [25,54]. Therefore, CBD should be considered as a multitarget agent, capable of triggering various scenarios, depending on the cellular and microenvironmental context, which includes a characteristic pattern of CBD-binding receptors, cellular metabolic state, CBD concentration, and bioavailability.

The involvement of CB1/CB2 receptors was addressed in several cancer models (Table A1). Specific antagonists of CB1 receptors or CB1 receptor knockdown abolished the antiproliferative effects of CBD in a colorectal cancer cell line [36]. Both CB1 and CB2 receptors were shown to be involved in the development of different processes induced by CBD, including autophagy in human neuroblastoma [44], proliferation and viability decrease in breast cancer [9], apoptosis in glioblastoma [18] and colon carcinoma [27], and reversed invasiveness of human cervical and lung cancer cell lines [39,40]. The involvement of CB2 but not CB1 in CBD-triggered effects was demonstrated in several models: the inhibition of proliferation and viability in murine thymoma and human leukemic cells [31] and human glioblastoma [12] and PARP cleavage in prostate carcinoma [27].

On the other hand, U87 and U373 human glioma cell lines [12,13] and glioma stem-like cells [16] express CB1 and CB2 but the antiproliferative effect of CBD was insensitive to the respective antagonists SR141716 and SR144528. CBD decreased the cell viability of D425 and D283 medulloblastoma and IC-1425EPN and DKFZ-EP1NS ependymoma cell lines, independent of CB1, even though human medulloblastomas and ependymomas express CB1 and CB2 [45]. CBD (5 µM) decreased the survival of the MDA-MB-231 breast cancer cell line in a CB1/CB2-independent manner [22], although the antiproliferative effect was partially CB2-dependent when CBD was added at a higher concentration (10 µM) [9].

The antitumor effects of CBD were shown to depend on TRPV1 in human neuroblastoma [44], cervical, lung, and breast cancer [39,40], and colon adenocarcinoma [36], but not in glioblastoma [12,13], human leukemia, or murine thymoma [31]. The CBD-dependent decrease in the viability of glioma stem-like cells was dependent on both TRPV1 and TRPV2 [16]. High expression of TRPV2 in drug-resistant cancers such as triple-negative breast or advanced non-small cell lung cancers is correlated with better prognosis, and the activation of TRPV2 by CBD assists drug (doxorubicin)-induced apoptosis in breast cancer cells or provokes apoptosis by CBD itself in lung cancer cells [55,56]. Reported results should be interpreted with caution, due to the different culture conditions, CBD concentrations, or mode of CBD application. For example, viability and proliferation in the MDA-MB-231 cell line (breast cancer) was reported to be independent of TRPV1, when cells were cultured in serum-free conditions [22]. A contradictory result was obtained in another study, where the same cellular model was used, but CBD was added daily in a similar concentration and cells were cultured with serum [9].

It should also be noted here that, in most works on the dependence of CBD-triggered processes on plasma membrane receptors, only late (12–48 h) events were studied (Table A1). For instance, CBD (5 μM, serum-free medium) produced multiple cytotoxic effects in Jurkat cells at 24 h, which were CB2-dependent [31]. Contrary to this, applied to the same cell model, CBD (30 μM, with serum) provoked almost instantaneous [Ca^2+^]_i_ and [Ca^2+^]_m_ rises and induced cell death, which were independent of CB1/CB2 and GPR55, but dependent on the direct modulation of the mitochondrial VDAC, [Ca^2+^]_m_ overload, and mitochondrial damage [25]. Although CBD-dependent ROS production has been confirmed in numerous cancer models, and CBD cytotoxicity is suggested to be related to oxidative stress, the question of the dependence of ROS production on any kind of plasma membrane receptors has not yet been addressed.

Proliferator-activated receptor γ (PPARγ) in peroxisomes appears to be an additional intracellular target for CBD. The antiproliferative effects of CBD (10 µM) in colon cancer cells were counteracted by PPARγ antagonist GW9662 [36].

An important issue is the selectivity of the drug for cancer vs. healthy tissues. Several pre-clinical studies demonstrated that the concentrations of CBD that were cytotoxic in cancer cell lines did not significantly decrease the viability of healthy cells such as primary glial culture (up to 50 µM CBD) [14], human oral keratinocyte cell line (up to 15 µM for 24 h) [50], human keratinocytes, rat preadipocytes and mouse monocyte–macrophage cell lines (10 µM for 72 h) [9], murine bone marrow stromal cells, and resting but not activated human CD4^+^ lymphocytes (30 µM for 24 h) [25]. The MCF-10A mammary epithelial cell line was more resistant to CBD than the MDA-MB-231 breast cancer cell line (up to 10 µM for 24 h) [22]. In some cases, apoptosis mediated by CBD (16 µM) occurred earlier in cancer (EL-4 murine thymoma, 1 h) than in healthy tissue (thymocytes, 6 h) [33].

Although, in general, healthy tissue cells seem to be less sensitive to CBD cytotoxicity, some of their functional properties may be affected. Murine splenocytes stimulated for cytokine production showed lower production of IL-2, IL-4, and IFN-γ after being pretreated with CBD [57,58,59]. On the other hand, the functionality could be restored later. For example, human resting CD4^+^ cells, pretreated with CBD (30 µM, 24 h), completely restored their ability to be activated after 72 h in CBD-free conditions [25].

The anticancer properties of CBD were confirmed in experiments in vivo. CBD reduced tumor growth and metastasis in animal models such as human xenografts of squamous carcinoma [50], colorectal and gastric cancer [35,37], lung cancer [39,40], prostate carcinoma [29], glioma [12], and neuroblastoma [43], as well as orthotopic implants in mice, such as medulloblastoma/ependymoma [45], breast cancer [21,23], and leukemia [31]. Working CBD doses were within 1–100 mg/kg body weight, typically 5 mg/kg [9,39,40,50], which is roughly equivalent to a low micromolar range. Administration of CBD daily or every 3 days caused a significant reduction of tumor volume and a strong reduction in metastatic spread to other organs, as well as the induction of apoptosis [31], inhibition of EGF/EGFR signaling [23], and activation of caspase 3 [43].

## 3. Synergism: CBD Improves the Effect of Conventional Anticancer Therapy

Synergistic effects of cannabinoids with other compounds were observed and discussed in early studies during the late 1990s, suggesting that CBD and other molecules such as terpenoids from *Cannabis sativa* can boost the activity of other compounds such as THC. This effect was denominated as the entourage effect; however, the evaluation of the synergistic effect of CBD with other drugs was mainly restricted to research on neurological diseases [60,61]. Eventually, the synergistic potential of CBD in other pathologies, including cancer, gained interest and became a subject of ongoing research.

Combined chemotherapy is the main therapeutic anticancer strategy that potentially reduces drug resistance. Accordingly, the search for the best drug combination is paramount. In this regard, the synergism of CBD with several cytotoxic drugs, including THC and conventional chemotherapeuticals such as gemcitabine, cytarabine (ARA-C), cyclophosphamide (CPA), cisplatin (CIS), doxorubicin (DOX), paclitaxel, temozolomide, carmustine, vincristine (VIN), carfilzomib, and erastin, as well as irradiation, was observed (Table 1) [10,11,15,45,62,63,64,65,66,67,68,69,70]. The synergistic effect was manifested either by an increase in cytotoxicity in vitro or by a decrease in tumor size in xenograft models. In multiple studies, the synergistic effect was quantitatively analyzed by the evaluation of the so-called Combination Index, CI, which has to be <1 in the case of synergy for a two-drug combination [71]. For a combination of CBD with different anticancer drugs, CI ranging from 0.22 to 0.9 was reported (Table 1).

In several cases, the combined effect of CBD with anticancer agents was non-trivial. In the study by Deng and colleagues [66], CBD itself exhibited pronounced cytotoxicity against several glioblastoma cell lines (with IC_50_ = 3.2 μM). However, synergism was demonstrated only when low CBD concentrations were combined with DNA-damaging agents, but not with the most of other drugs, where the effect of drug combinations was only additive or even antagonistic [66]. Another study demonstrated that CBD synergistically enhanced the cytotoxic effects of CPA in different MDB cell lines, but only at high concentrations, whereas low CBD concentrations (<5 μM) antagonistically interfered with the CPA activity [45]. Tamoxifen (TAM) was shown to interact synergistically with CBD in the suppression of T-ALL cells and this synergism was higher when cells were pretreated with TAM or both drugs were added simultaneously compared to the case of TAM after CBD. This was explained by the fact that TAM pretreatment prevented the mitochondrial permeation transition pore formation by binding to cyclophilin D, so that a consequent CBD application resulted in a permanent mitochondrial Ca^2+^ overload and more severe mitochondrial dysfunction [72].

The outcome of CBD interactions depends on the cancer type/phenotype and microenvironmental conditions. For example, non-identical interactions of CBD with other cytotoxic drugs (e.g., CPA, THC) were observed in two medulloblastoma cell lines: synergism in D283 and antagonism in PER547. Remarkably, the synergism observed in D283 in vitro was not confirmed in the xenograft environment [45]. Similarly, CBD acted synergistically with ARA-C in myeloid leukemia cells but not in acute lymphoblastic leukemia [10]. Another issue is the nature of the companion anticancer agent. In myeloid leukemia, CBD exhibited synergistic effects with ARA-C, but antagonism with VIN [10].

Thus, the possible outcome of CBD interaction with any of the conventional chemotherapeutic agents should be carefully examined in its context, which includes the experimental model, the cancer phenotype, the nature and the concentration of the drug, as well as individual patients’ particularities.

## 4. CBD in Palliative Care

Standard anticancer treatments such as chemotherapy, radiotherapy, hormone therapy, and nutritional adaptations are known to impact negatively on the patients’ life quality by disrupting sleep and appetite, producing pain, increasing the appearance of mood disorders, and generating immunosuppression, anemia, fatigue, and multisystemic toxicity, especially with intensive or long-term protocols. In this context, there is a great deal of interest in palliative care [73]. Despite the general popularity of the topic, there is only a limited number of published studies regarding the palliative properties of cannabinoids, which present significant methodological flaws. We will discuss some of these reports in more detail.

More than 3000 cancer patients using medical cannabis were monitored for 2 years in Israel to assess its safety and efficacy [74]. Of these patients, 66% reported a substantial improvement in their health condition and life quality from the first month of use. Despite the fact that the obtained results are encouraging, several limitations complicate their interpretation: (1) the medicament formulation was not of pharmaceutical purity grade and represented whole plant oil extract or inflorescence, including flowers, capsules, or cigarettes; (2) data from all patients were combined and analyzed regardless of patient age, cancer type, and stage. In another study from the Mayo Clinic published recently, patients, including cancer patients, used THC and CBD as a palliative agent against pain, appetite loss, and insomnia [75]. In the majority (71%) of patients using CBD, these symptoms were alleviated. However, there were many uncontrolled variables in the CBD consumption, including concentrations (not reported), frequency of consumption (daily, weekly, or rarely), and methods of administration (vape, spraying, pills, topical application). In the majority of trials so far, instead of pure CBD, CBD/THC formulations with different ratios and purity were employed.

It is worth mentioning here that the consumer demand for CBD products has increased drastically during the last decade [76]. As a result of this rising demand, numerous CBD-containing products have appeared for online purchase. In a recently published study, eighty-four CBD products from 31 companies were analyzed for whole-spectrum cannabinoid content (CBD, THC, cannabinol, cannabigerol, among others) using high-performance liquid chromatography [77]. CBD concentrations varied significantly, from 0.10 to 655 mg/mL, with only 31% of accurately labeled products. The rest of the products were either underlabeled (43%) or overlabeled (26%). Mislabeling occurred frequently in vaporization liquids. Importantly, THC was detected in 20% of samples, sometimes in concentrations sufficient to provoke intoxication. These findings indicate the urgency of manufacturing control and testing standards, to prevent inappropriate use. In this context, double-blind, placebo-controlled, randomized clinical trials to assess the use, efficacy, and safety of CBD in palliative care are now being conducted [4,78].

### 4.1. CBD in Chemotherapy-Induced Pain

It is estimated that around 70–90% of patients with advanced cancer experience pain during therapy due to the therapy-induced damage in the peripheral nerves. There is an extensive search for strategies to limit the development of chemotherapy-induced neuropathic pain (CINP) or to relieve pain, in order to improve patients’ life quality [74,79,80].

CBD has been demonstrated to exert analgesic effects in a murine model of CIS-induced allodynia [81]. Similar effects were also observed in cancer patients, followed for up to 6 months of CBD consumption, with a significant reduction in pain caused by chemotherapy. Most of the patients (67%) stopped using analgesics or reduced the dosage [74].

There are at least 76 clinical trials, either completed or recruiting, which evaluate the benefits of CBD in pain management. Among them, 17% are focused on the analgesic properties of CBD in cancer patients (www.clinicaltrials.gov, accessed on 14 February 2022). Such trials employ CBD either alone or in combination with other cannabinoids in doses ranging from 2.5 mg to 40 mg, mostly via an oromucosal spray. Low (<25 mg) doses of CBD provoked analgesia, while higher doses caused no analgesia but secondary effects [82]. Patients with terminal cancer-related pain and refractory to opioids experienced a decrease in pain severity within the first few weeks of CBD/THC consumption [83,84]. Similar results were obtained in another trial, in which more than 30% of patients reported a reduction in baseline pain [83]. Contrary to these findings, several independent double-blind, placebo-controlled phase 3 trials showed no significant difference between CBD/THC and placebo effects on pain management, although patients reported some improvement in their life quality (NCT01361607; NCT01424566) [85,86].

Based on data from in vitro and clinical trials, several mechanisms have been proposed for CBD-mediated analgesia, which includes the action through different cell membrane receptors, ion channels, transporters, as well as intracellular enzyme targets [8]. However, there are only a few studies on tumor models. Accordingly, in a breast cancer xenograft, CBD (2.5–10 mg/kg) prevented the manifestations of CINP induced by paclitaxel, acting through the serotonin receptor 5TH1A [64]. Importantly, CBD treatment also displayed a synergism with paclitaxel against breast cancer cells [64]. In another work, CBD (0.625–20 mg/kg) was shown to attenuate CINP, induced by paclitaxel or oxaliplatin, but not by vincristine [87]. Additional experiments are still needed to confirm the analgesic effects of CBD on chemotherapy-induced neuropathic pain and to reveal the underlying mechanisms.

### 4.2. CBD for Healthy Cells’ Protection

Several anticancer agents are toxic to healthy cells, especially when the drugs are accumulated in certain organs. For example, it is well known that CIS promotes acute renal failure (ARF) in a dose-dependent manner in approximately one third of patients [88,89]. CIS is differentially absorbed by the medullar and cortical sections of the kidney, inducing apoptosis and necrosis in these tissues. Several mechanisms have been implicated in CIS-mediated nephrotoxicity; thus, drugs limiting such mechanisms have emerged as renoprotective agents [90]. In an ARF mice model, the pre-administration of CBD (10 mg/kg/day) significantly attenuated the renal damage induced by CIS [89]. Additionally, CBD has been shown to potentiate CIS activity in different cancer types [62,66]. Thus, CBD may be considered as a promising renoprotector against CIS-induced renal failure.

Another effective chemotherapeutic drug, DOX, may provoke cardiotoxicity when accumulated. For cancer patients with DOX-developed cardiomyopathy, the prognosis is poor [91,92]. In mice with DOX-induced cardiomyopathy, CBD (10 mg/kg/administrated i.p. for 5 days) reduced the markers of ARF and cardiac injury [93]. The effects of CBD as a cardioprotector were attributed to a reduction in oxidative/nutritive stress and cell death and it improved the mitochondrial function and biogenesis. From a therapeutic point of view, CBD usage in cancer patients under regimens including DOX is encouraged, considering that CBD also potentiates the cytotoxic effects of DOX (Table 1), allowing the adjustment of DOX doses and limiting its cardiotoxicity.

### 4.3. CBD against Opportunistic Infections

Cancer patients undergoing chemotherapy are at high risk for opportunistic infections. It is estimated that 30% of cancer patients with non-hematological tumors and up to 85% of patients with acute leukemia develop life-threatening infections. Some chemotherapeuticals such as CPA cause immunosuppression by altering hematopoiesis, affecting the total white blood cell count and generating neutropenia [45,94,95]. Chemotherapy, surgical, or diagnostic procedures can also disrupt anatomic barriers in the process of infection. To overcome these complications, the concurrent use of antimicrobial agents and growth factors to restore hematopoiesis is being considered [96].

In this regard, the ability of CBD to influence hematopoiesis was observed. For example, in orthotopic mouse models of ependymoma and medulloblastoma, CBD (50 mg/kg/p.o.) was able to reverse hematopoietic toxicity caused by CPA treatment, as measured by an increase in leukocyte and neutrophils counts. However, the survival rate of animals in this model was not improved, despite the fact that, in experiments on medulloblastoma and ependymoma cell lines performed in vitro, CBD enhanced the cytotoxic effects of CPA (Table 1) [45]. Notably, an increase in the total number of white blood cells, lymphocytes, monocytes, and neutrophils was also seen in cannabis consumers [97]

Several studies evidenced the marked antimicrobial activity of CBD. In particular, CBD was effective against various species of Gram-positive bacteria, including *Staphylococcus* spp., *Listeria* spp., *Enterococcus* spp., and *Bacillus* spp., with the range of minimum inhibitory concentrations (MIC) being 1–4 μg/mL [98,99,100,101,102]. CBD also potentiated the effect of bacitracin [101]. Importantly, it was highly efficient against many Gram-positive resistant strains [102]. Although the majority of Gram-negative species are significantly less sensitive to CBD (MIC > 60 μg/mL), some “urgent threat” pathogens such as *Neisennia honorrhoeae*, *Neisennia meningitides*, and *Legionella pneumophilia* showed high sensitivity, with MIC around 1 μg/mL [102]. In addition to its bactericidal properties, CBD protects the mucous membrane and limits the susceptibility to infections due to its antisecretory, antioxidant, anti-inflammatory, and vasodilatory properties [103,104,105,106].

### 4.4. CBD in Anorexia-Cachexia Syndrome

Up to 80% of cancer patients undergo a wasting syndrome, characterized by vomiting, anorexia, asthenia, and anemia [107,108]. The resulting cancer cachexia (CCA), together with immunosuppression, increases their susceptibility to infections, limits chemotherapy’s effectiveness, and increases the risk of eventual organ failure. Therefore, cancer patients are strongly encouraged to adopt strategies that promote appetite increase, weight gain, and immunity recovery. Advanced cancer patients treated with CBD (2.5 mg p.o.) or CBD/THC blends showed improved appetite compared to the placebo group [84]. Another controlled study confirmed weight gain in cancer patients receiving CBD of pharmaceutical grade (20 mg/daily/p.o.) [109]. Interventional phase 2/1 clinical trials have been proposed in order to evaluate the effects of CBD in emesis, cachexia, and appetite alterations by estimating the body mass index, nausea, taste alteration, energy intake, and lean body mass in cancer patients under chemotherapy (NCT03245658; NCT04585841; NCT04482244; NCT02675842).

Collectively, available data suggest that CBD can improve the life quality of cancer patients under chemotherapy (Figure 2) and call for further extended clinical trials of CBD as a potential palliative care agent.

## 5. Evidence of Anticancer Activity of CBD from Clinical Trials and Case Reports

Although numerous pre-clinical studies have demonstrated the anticancer activity of CBD (Section 2 and Section 3), objective clinical evidence is still very scarce. A comprehensive review of pre-clinical and clinical reports concerning the anticancer activity of cannabinoids, including CBD, was performed and published recently [110]. In this work, the data available in the PubMed and EBSCO databases, congress presentations, books, and clinical trials registered at ClinicalTrials.gov website were analyzed. Among them, 77 publications of case reports with various types of cancers were revealed and classified as weak (81%), moderate (5%), or strong (14%). Accordingly, the cases were considered as strong or moderate when they met the following criteria: (a) patients presented an active form of cancer at the time of cannabinoid application and (b) clinically validated laboratory documentation about clinical response and improvement was available. In strong cases, cannabinoids were utilized without a concurrent therapy, whereas in moderate cases, anticancer therapies were executed in parallel. In our opinion, the latter combined approach is more pertinent than CBD monotherapy. In clinical trials reported by Kenyon and colleagues, pharmaceutical-grade synthetic CBD (STI Pharmaceuticals) was tested on 119 patients with advanced cancer of different types, including breast, prostate, and colorectal cancers, non-Hodgkin’s lymphoma, and glioblastoma [109]. Patients were given 10–30 mg of CBD (depending on tumor mass), twice per day (“three days on/three days off” basis). Favorable clinical responses were observed in 92% of patients, evidenced by a reduction in tumor size (repeated scans) and a decrease in circulating tumor cells. Positive dynamics were observed in patients treated with CBD both alone and in combination with a standard therapy [109]. Importantly, the authors reported the case of a glioma patient where the improvement was observed only by taking synthetic CBD of pharmaceutical grade, but not cannabis oil extract [109]. It should be noted here that clinical researchers, physicians, and the FDA expressed their concern that many patients use a variety of cannabis oils or whole plant extracts of questionable quality (not of pharmaceutical grade) in self-prescribed dosages, which may be ineffective or even harmful for patients [109,111].

Thus, the following important issues should be addressed in the path toward the use of medical CBD for cancer patients: (1) CBD formulations and administration methods to reach the desirable cytotoxic effect specifically in the cancer tissue or favorable effects in palliative care; (2) possible side effects for specific CBD formulations and concentrations administrated by any specific route.

## 6. CBD Tolerability, Toxicity, and Adverse Effects

CBD’s toxicity against numerous cancer cell lines has been identified, as previously discussed (Section 2). Although healthy cells have been reported to be less sensitive, the causes and mechanisms of the differential sensitivity of cancer and healthy cells to CBD toxicity are still unclear. Moreover, CBD may target a variety of surface and intracellular molecules (receptors, ion channels/transporters, enzymes) and triggers multiple signaling pathways present in both cancer and healthy cells. Taken together, these facts raise safety and side effect issues. According to traditional protocols, drug toxicity is first tested in pre-clinical animal models. Pre-clinical studies, carried out on animal models, reported acute and chronic adverse effects of CBD on different organs and systems (Table A2) [112,113,114,115,116,117,118,119,120,121,122]. There are several highly recommended comprehensive reviews, which critically analyzed the CBD safety and toxicity experiments carried out in animal pre-clinical and human clinical trials [53,123,124,125,126,127]. The following important observations should be mentioned: (1) regarding the administration route, in most human trials, CBD was administrated orally or by inhalation, whereas predominantly intraperitoneal (i.p.) and intravenous (i.v.) injections and sometimes the oral route were used in animals; (2) CBD pharmacokinetics and molecular targets seem to differ between humans and rodents; these differences should be taken into consideration when extrapolating results obtained in pre-clinical models to humans; (3) regarding the composition, in numerous CBD toxicity reports in humans, patients consumed not pure CBD but different preparations of CBD of unknown concentration and uncertain composition. Many preparations marketed as CBD contain also variable quantities of THC [77]. Since the toxicity profile and side effects caused by THC and CBD are different and THC seems to be more toxic [126], the data obtained in these studies are misleading, reporting the net effect of THC, CBD, and their interaction. Drug–drug interactions represent a very important issue in the case of CBD, because it targets enzymes implicated in drug metabolism and excretion [8]. Thus, it may prolong the presence and increase the toxicity of co-administrated drugs. Taking all the aforementioned factors into consideration, we will restrict ourselves to the most prominent and reliable data concerning the toxicity and adverse effects of CBD.

Obviously, CBD’s tolerability depends on the doses, frequency, routes of administration, and treatment duration. CBD is usually well tolerable during acute and short-lasting treatment in moderate doses. At a range of 3–30 mg/kg (i.p.) or 0.1–30 mg/kg (i.v.), CBD did not change the heart rate, blood pressure, gastrointestinal (GI) transit, respiration, biochemical blood parameters, and hematocrit in rodents [53]. In piglets, CBD doses of 10 mg/kg (i.v.) were well tolerated, whereas higher doses (50 mg/kg) in some cases caused hypotension and cardiac arrest [116,125]. In rhesus monkeys, high CBD doses of 150–300 mg/kg (i.v.) caused acute CNS toxicity (tremor, sedation, and prostration) within 30 min of injection, whereas prolonged treatment for 9 days elicited bradycardia, hypopnea, cardiac failure, liver weight increase, and inhibition of spermatogenesis [113,125]. For the same model (rhesus monkeys), chronic oral CBD application (30–300 mg/kg/day, 90 days) caused systemic negative effects on the liver, heart, kidneys, and thyroids, and inhibited spermatogenesis [113,124,125]. Negative effects of chronic CBD on embryonic development were reported in rats when relatively high doses (75–250 mg/kg/day) were administrated orally during pregnancy, which included developmental toxicity, decreased fetal body weight, increased fetal structural variations, and embryofetal mortality [125].

Clinical reports in humans are scarce, and, obviously, are limited to low and moderate doses. No disturbances in physiological parameters or psychomotor functions were observed in clinical CBD trials after oral administration (15–160 mg), i.v. injection (5–30 mg), or inhalation (0.15 mg/kg) [53]. No side effects were observed during the prolonged CBD treatment of cancer patients (up to 60 mg daily, orally, up to 6 months) [109].

Most of the reliable clinical trials (i.e., double-blind, randomized, placebo-controlled) were performed on patients (children and adults) suffering from treatment-resistant epilepsy or schizophrenia, or related neurologic and psychotic disorders. The CBD dose range utilized in these trials was usually from 0.5 to 50 mg/kg/day or from 200 to 1000 mg/day for psychiatric studies. When CBD was administrated orally (25–50 mg/kg/day) for an extended period (weeks), moderate adverse effects included somnolence and fatigue, sleep disorders, diarrhea and GI intolerance, and respiratory complications, and pneumonia, thrombocytopenia, and liver and blood abnormalities were reported [125,128,129,130,131]. Pyrexia was relatively common in children with Dravet’s or Lennox-Gastaut syndrome during 3- or 4-week treatment trials with doses of 5–20 mg/kg/day administered orally [125,128,129].

Since CBD is suggested to be included in combined anticancer chemotherapy protocols, CBD’s hepatotoxicity, which can cause changes in drug metabolism, is an issue of special importance. A hepatotoxic effect was documented in pre-clinical and clinical studies when relatively high CBD doses were administrated for a prolonged time [53,123,124,125,126,127]. As was revealed by a randomized, double-blind trial that included 171 patients, hepatocellular injury represents the most frequent adverse effect, so it was recommended to test serum transaminases and total bilirubin levels in all patients prior to starting the treatment with Epidiolex^®^, which is CBD in an oral solution [132,133]. Importantly, CBD targets the cytochrome P450 system and is metabolized by CYP3A4 and CYP2C1 in human liver microsomes (HLMs), giving rise to 6α-OH-, 6β-OH-, 7-OH-, and 4″-OH-CBDs [134]. A female patient, treated for 6 years with tamoxifen, and, additionally, by CBD, which inhibited CYP3A4/5 and CYP2D6, presented a consequent reduction in *N*-desmethyltamoxifen and active metabolite endoxifen [135]. In cancer patients, especially if they have liver diseases or a poor metabolic profile, possible effects of CBD on cytochromes P450, which in turn can affect the pharmacokinetics of conventional anticancer drugs, need to be considered.

## 7. Concerning Better CBD Delivery for Cancer Therapy

Satisfactory delivery of anticancer therapeuticals should provide its efficient accumulation in the target cancer tissue, with minimal side systemic effects on other organs. CBD is a highly lipophilic compound, which is poorly soluble in aqueous solutions and highly sensitive to light, temperature, and oxidation, which underlies its relatively low bioavailability [136]. CBD, when administrated orally, can precipitate in the GI tract, resulting in poor GI permeability. It undergoes then the first step of metabolism by liver and gut enzymes and is predominantly excreted through the kidneys [136,137]. As a result of the first step of metabolism, the oral CBD bioavailability is estimated to be between 5% and 19% [136,137]. Variable pharmacokinetics profiles were reported, depending on the means of CBD administration. These include more traditional and better-studied oral/mucosal, inhalation, and smoking, and less explored intravenous routes [138].

### 7.1. Free CBD Delivery

To date, the only CBD formulation approved by the FDA for the treatment of rare forms of epilepsy is Epidiolex^®^, CBD in an oral solution (100 mg/mL), with maximum recommended doses of 20 mg/kg/daily. Currently, there are numerous clinical trials of CBD for the treatment of different disorders, including palliative care in cancers, where CBD is delivered predominantly as an oil solution, orally, or via inhalations (https://clinicaltrials.gov/ct2/results?cond=&term=cannabidiol&cntry=&state=&city=&dist=, accessed on 14 February 2022).

As was discussed in Section 2, a relatively broad range of CBD concentrations was tested in studies in vitro to prove its anticancer properties. Significant variations in experimental models and culture conditions complicated a comparative analysis. Considering cell cultures supplemented with serum as a better approximation of physiological conditions, effective concentrations were in the μM range. When CBD was administrated orally in humans (20 mg), its maximal plasma concentration achieved at 3 h was in the range of 7.9–19.1 ng/mL (i.e., 5–15 nM), with better bioavailability in women (Table 2) [139]. A novel self-emulsifying drug delivery system (SEDDS) was proposed recently to improve the oral CBD bioavailability. This resulted in 2–4-fold higher plasma CBD concentrations when compared to oral/mucosal administration, with a lower gender difference (Table 2) [139,140,141]. When CBD was received by inhalation or smoking, the maximal plasmatic CBD levels were in the nM range and then dropped stepwise (Table 2) [142,143].

The first trial in adult humans to compare single and multiple oral delivery was undertaken recently [144]. The single oral dose was administrated in the range of 1500–6000 mg, which is comparable to or higher than doses recommended for Epidiolex^®^. The maximal plasma CBD concentration, reached at 3–5 h after administration, was 292.4 ± 87.9 ng/mL (approx. 1 μM) and 782 ± 83 ng/mL (approx. 2.5 μM) for 1500 and 6000 mg, respectively, and then it dropped significantly. When CBD was administrated twice per day (2 × 1500 mg) during an extended period of 7 days, a steady-state plasma level was reached at 2 days, and on day 7, the maximal concentration was 541 ng/mL (approx. 1.7 μM).

Intravenous CBD injection is an alternative delivery method, which prevents GI degradation and has demonstrated better bioavailability. It was tested and compared with other delivery methods in studies in humans and mice (Table 2) [142,143,145]. Intravenous administration caused higher CBD plasma levels than oral administration [145], smoking [142], or inhalation [143] (Table 2). In healthy volunteers, the injection of a 20 mg/kg dose resulted in a rapid rise in the plasma concentration, ranging from 358 to 972 ng/mL (1–3 μM), which was approximately five times higher than by smoking [142]. Although these concentrations are close to the cytotoxicity range reported for some tumors (discussed in Section 2), plasma CBD levels had dropped drastically within 1 h of administration [142]. Similar results were obtained in a murine model, with an immediate plasmatic concentration rise to 3000 ng/mL (approx. 10 μM), when 10 mg/kg was injected, followed by a rapid (within 1 h) tenfold drop [145].

**Table 2 pharmaceuticals-15-00366-t002:** Comparative studies of alternative routes of free CBD administration.

Participants	Delivery Method Doses	Plasma Concentration, ng/mL	Reference
Young healthy male volunteers (*n* = 5)	Smoking 20 mg	Max at 3 min: 110 ± 55 Max at 1 h: 10.2 ± 6.6	[142]
	i.v. 20 mg	Max at 3 min: 686 ± 239 Max at 1 h: 48.4 ± 10.7	
Male ICR mice (*n* = 3)	p.o. 20 mg/kg	Max at 2 h: 111 ± 52 Max at 4 h: 60 ± 58	[145]
	i.v. 10 mg/kg	Max at 10 min: 3343 ± 1048 Max at 1 h: 376 ± 229	
Healthy male/female volunteers (*n* = 8/8)	p.o. 25 mg	Max at 3 h: 3.05: range: 1.57–4.54 Max at 8 h: 1	[139]
	p.o., SEDDS 25 mg	Max at 1 h: 13.53, range: 7.9–19.1 4 h: 2.5	
Healthy male/female volunteers	inhalation, THC/CBD 20/20 mg	5 min (max): 2–17	[143]
	i.v., THC/CBD 10/10 mg	Max at 5 min: 14–26	
Healthy male/female volunteers	p.o., single dose 1500 mg 3000 mg 6000 mg	Max at 5 h: 292.4 ± 87.9 533.0 ± 35.1 782.0 ± 83.0	[144]
	p.o., multiple dose 2 × 750 mg or 2 × 1500 mg daily	Max at 7 d: 330 541	

Thus, any administration route of free CBD resulted in a transient rise in the plasmatic drug level, where only the maximal levels are comparable to cytotoxic concentrations. Importantly, bioavailability in cancer tissue is expected to be significantly lower than in plasma and highly variable, depending on the cancer type, tumor size, geometry, and vascularization. On the other hand, achieved plasma concentrations are sufficient to cause undesirable side effects (Section 6). Thus, increasing the dose of pure CBD by any administration method should not be considered as an appropriate strategy for CBD delivery for cancer treatment. Instead, alternative formulations, aimed to increase CBD’s stability and its specific targeting to the cancer tissue, should be developed.

### 7.2. Nanotechnology May Improve CBD Delivery for Cancer Therapy: General Considerations and Experimental Evidence

Multiple nanoformulations have been proposed to overcome the delivery challenges of hydrophobic unstable drugs such as CBD. There are several excellent comprehensive reviews discussing in detail the best approaches to design nanocarriers (NC) for cancer therapeutics [146,147,148]. There are various important criteria that should be taken into consideration. NC should be composed of biocompatible nontoxic and non-immunogenic materials. According to their chemical structure, NP can be categorized into different groups, such as inorganic, polymeric, liposomas, nanomicelles, etc. In inorganic nanoparticles, the core is composed of metal or metal oxide (silver or gold are frequently used). Polymeric NC are produced using a conjugation of several polymers with desirable characteristics. Liposomes are nanoparticles with an aqueous interior part, surrounded by one or more concentric bilayers of amphipathic lipids (e.g., phospholipids). The design of such NC can be developed according to therapeutic requirements. Their diameter ranges normally from 1 nm to several μM. Consequently, such liposomes can be distributed in the bloodstream (smallest capillary diameter is approximately 5–6 μM) and accumulated in the target tumors. The ultra-filterable range of less than 200 nm provides the possibility for sterilization. Covalent linkage of NC to polyethylene glycol (PEG), so-called PEGylation, decreased significantly their immunogenicity. Moreover, such a modification changes the physicochemical and hydrodynamic properties, which results in a prolonged circulation time and reduced renal clearance [149]. NC easily incorporate drug molecules and form a barrier around therapeutic agents, preventing the premature drug interaction with body fluids and immune cells before their delivery to the target site. A precise design, which takes into consideration the material, size, and shape of NC, may provide drug release in a controlled and predictable fashion. This approach is also useful for the delivery of two or more drugs simultaneously, which can be very useful for cancer treatment, considering multi-drug chemotherapeutic protocols. Moreover, the nature of the core molecules may provide the possibility to combine both hydrophobic and hydrophilic drugs at the same time. In liposomes, hydrophobic drugs are incorporated into the lipid membrane, whereas hydrophilic compounds are present within the central aqueous cavity.

Target-specific drug delivery can significantly decrease side effects and increase the therapeutic index of encapsulated drugs. Passive and active targeting of nanoparticles can be used for cancer therapy. Passive targeting is possible due to the phenomenon known as the enhanced permeability and retention (EPR) effect in solid tumors [147,150,151,152,153]. In rapidly growing tumor tissue, characterized by the overexpression of vascular endothelial growth factor (VEGF), the microvasculature is characterized by a chaotic ramification with enhanced endothelial porosity or fenestration, in contrast to the tighter endothelial structures of normal capillaries. As a result of the changed cytoarchitecture, the blood flow is slower, and, due to the high porosity, tumor capillaries are leaky. Both these factors ensure the retention of enlarged particles, such as NC, in tumors. In hematological malignances, the bone marrow (BM) leukemic niche is the target tissue. Blood vessels supplying BM (sinusoids) possess the fenestrations and are semipermeable, providing favorable conditions for the accumulation of NC [154]. At the same time, the EPR effect was reported to provide a relatively modest, twofold enhancement of the nanodrug retention in tumor tissues, when compared with healthy organs [155].

The surface of NC can be modified to improve their targeting to tumors. A variety of ligands/antibodies to specific antigens, expressed by cancer cells, can be proposed for NC surface engineering [146]. Dual-action CXCR4-targeting liposomes were developed and proposed for drug delivery and the simultaneous blockage of the CXCR4/CXCL12 axis for leukemia treatment [156]. HER2-targeted liposomes were accumulated in the tumor tissue of patients with HER2-positive breast cancer [157]. The RGD (arginyl/glycyl/aspartic acid) motif was proposed to target integrins to tumor cells [158]. Anionic liposomes were shown to accumulate in BM and were then predominantly adsorbed by leukemic cells [154]. Hyaluronic acid, which shows a high binding affinity for the CD44 adhesion molecule, is present at enhanced concentrations in a variety of tumors and was also proposed for NC modification [159,160]. Experimental trials of novel delivery methods for CBD in cancer therapy are still scarce but have demonstrated promising results (Table 3) [161,162,163,164,165,166,167,168].

Gold PEGylated nanodrones were proposed recently to target lung cancer with cannabinoids and radiosensitizers [161]. The efficiency of two administration routes, inhalation and intravenous, was tested in transgenic mouse models bearing lung adenocarcinoma. The particle size (100 nm) was optimized to ensure an increased circulation time and efficient tumor uptake. Additionally, drones were functionalized with the RGD (arginyl/glycyl/aspartic acid) motif to target integrin receptors on the lung tumor cells’ surface. Both administration routes provided efficient nanodrone penetration into the tumor tissue, but the inhalation route was more promising for this tumor type. CBD was proposed to be conjugated to the amine groups present on the PEG. However, CBD-conjugated drones have not been tested yet.

The efficiency of a micellar delivery system for targeting cannabinoids to cancer tissue was tested in a murine model of triple-negative breast cancer [162]. In this case, micelles were loaded with the synthetic cannabinoid WIN55,212-2. The average micelle size was 152 nm, ensuring their accumulation in the tumor by the EPR. WIN, being conjugated with the micellar system, efficiently inhibited tumor growth. Remarkably, predominant micelle accumulation in the tumor was demonstrated, indicating the viability of the micellar system for its use with cannabinoids.

CBD-loaded poly-ε-caprolactone microparticles, as an alternative delivery system for long-term CBD administration, demonstrated their efficiency in inhibiting glioblastoma growth and tumor angiogenesis in a murine xenograft model [163].

More recently, poly-(lactic-co-glycolic acid), PLGA, microparticles, loaded with CBD, were tested for their potential to improve the conventional chemotherapy of breast and ovarian cancers [164,165]. PLGA is approved by the FDA for use in parenteral release systems. The mean particle size was around 25 μM, with a high entrapment efficiency in the tumor tissue. Particles were sterilized by gamma irradiation (25 kGy). Since sterilization accelerates the polymer erosion, a CBD:polymer ratio (10:100) was selected to ensure a durable release profile. Remarkably, a single administration of this formulation ensures the antitumor activity in vitro for at least 10 days. CBD-loaded microparticles were effective as a monotherapy, but synergism with DEX (breast cancer) and paclitaxel (breast and ovarian cancer) allowed a more pronounced effect at a single administration. However, a particle size in the μM range is not suitable for intravenous injections, because only particles smaller than 5 μM can freely circulate in the bloodstream and reach the tumor site. Afterwards, PLGA CBD-loaded nanocarriers for i.p. administration in ovarian cancer treatment were developed, which demonstrated improved CBD stability, its long-lasting release, internalization by cancer cells, and anticancer efficiency [165].

Drug delivery to brain malignancies such as glioma/glioblastoma is restricted by the blood–brain barrier (BBB). Aparicio-Blanco and colleagues proposed the original strategy of non-immunologic BBB targeting using NC decorated (functionalized) with CBD [166,167]. They elaborated small lipid nanoparticles with a size range of 10–100 nm, carrying CBD on their surface, which were able to pass through the BBB. CBD-decorated particles were suggested to target the brain endothelium, which expresses different surface molecules able to bind CBD, namely the CB1 receptor, the G-protein-coupled receptor 55 (GPR55), and serotonin receptor 5-HT. After the brain endothelium transcytosis, these particles were expected to target glioma cells overexpressing CB1/2 receptors. As far as CBD was reported to be cytotoxic for glioma, lipid nanoparticles were loaded with CBD and tested as prolonged-release carriers for glioma therapy [166,167]. This strategy was demonstrated to enhance the glioma targeting, and a combination of CBD loading with CBD functionalization significantly reduced the IC_50_ values. CBD decoration was confirmed to enhance the passage of lipid nanoparticles across the BBB both in vitro (human brain endothelial hCMEC/D3 cells) and in vivo (mouse glioma xenograft models).

An RGD proteinoid polymer was synthesized and used to encapsulate CBD [168]. Resulting nanoparticles inhibited tumor growth in xenograft mouse models of colorectal and breast cancer and were proposed for further trials.

The possibility of the delivery of two or more drugs simultaneously by nanocarriers is of special interest for the inclusion of CBD into chemotherapeutic protocols, taking into the account the fact that CBD improves the effect of various anticancer drugs (Section 3). Importantly, there are several anticancer drugs that are already clinically used in liposomal formulations for chemotherapeutic protocols [169]. Among them are doxorubicin (Doxil^®^, 1995 and Myocet^®^, 2000), danourobicin (DaunoXome^®^, 1996), cytarabine (Depocyt^®^, 1999), mifamurtide (Mepact^®^, 2004), vincristine (Marquibo^®^, 2012), and irinotecan (Onivyde^TM^, 2015). Recently, pure CBD, encapsulated in a lipid bilayer for enhanced CBD delivery (liposomal CBD), was developed by InnoCanFarma (https://www.newsfilecorp.com/release/72614/Innocan-Pharma-Announces-Successful-Production-of-CBD-Loaded-Liposomes-under-Aseptic-Conditions, accessed on 14 February 2022).

## 8. Regulation Issues

Although CBD lacks any psychotropic effect, its public and clinical usage falls under general regulations applied to cannabis-derived products. Even though the difference between non-psychotropic components and THC is understood, recent circulars released by the U.S. Food and Drug Administration [170,171] strive to thoroughly evaluate the purity of cannabis products containing CBD and its derivatives and to inform the public about the risks and unknowns of these products. The current trend in several countries, including the U.S., Mexico, Canada, and Uruguay, to name those in the western hemisphere, is the decriminalization of the use of cannabis products. In 31 out of 45 European countries, CBD is legal or is within a grey legal zone (https://www.legalreader.com/cbd-in-europe-legal-status-of-cbd-country-by-country/, accessed on 10 March 2022). However, the regulations differ from country to country and even between different states. In Mexico, in 2020, an initiative was launched to differentiate between marijuana and non-psychoactive cannabis, and respective modifications were made in the General Health Law and Federal Penal Code, approved by the Chamber of Deputies in 2021. In the case of medicinal, palliative, pharmaceutical/cosmetic, or scientific uses for said purposes, these will be regulated by the provisions of the General Health Law and other applicable regulations. The Federal Commission for the Protection against Sanitary Risks (COFEPRIS) has publicly reiterated its “open door” policy to receive and guide all people and organizations interested in the medicinal, personal, and recreational use of cannabis. However, respective requests are attended individually and there is still a long way to go so that the requests do not necessarily have to be evaluated personally on a case-by-case basis, or by court order, but under general and clear guidelines and a regulatory framework that allows the definition of the therapeutic indications in which the use of cannabis-derived products will be prescribed, as well as the process to evaluate their quality, safety, and efficacy, in a similar way as occurs for other medicines. Therefore, researchers and clinicians who seek to employ CBD for anticancer treatments are strongly advised to consult the current status of respective regulations in their area.

## 9. General Conclusions and Further Considerations

The anticancer properties of CBD against cancer cells of different histogenesis were demonstrated in numerous pre-clinical in vitro studies (Section 2, Figure 1, Table A1). For many, but not all, cancer types, anticancer effects were also confirmed for animal models (Table A1, [6]). The synergic effect of CBD with conventional anticancer drugs encourages the inclusion of CBD in conventional chemotherapeutic protocols (Section 3, Table 1). The urgent need for clinical trials in developing CBD as an anticancer drug is proclaimed [6]. We provide here the suggested flowchart for the translation of the CBD anticancer activity from the lab to clinical trials and clinical use in anticancer treatments (Figure 3).

CBD acts through various molecular targets and triggers multiple signaling pathways simultaneously so that precise cytotoxic mechanism(s) for every cancer type are still to be revealed. Many studies evidenced lower, if any, cytotoxicity of CBD against healthy tissues, but the cause of the differential sensitivity of healthy and cancer tissues is still unclear. Moreover, the specific microenvironment may protect cancer cells from drug-induced damage. Thus, experiments in pre-clinical in vitro models with a high approximation of the cancer microenvironment, namely 2D and 3D co-culture with stromal cells and cancer organoids, are very desirable. The anticancer effects of CBD are often observed at relatively high concentrations of pure CBD added to cell culture or injected into animals, which can cause adverse effects, especially under long-lasting treatments (Section 6 and Section 7, Table A2 and Table 2). On the other hand, low CBD concentrations may even promote cancer cell proliferation [25]. Thus, new CBD formulations for targeted cancer treatments are required. NC of different design represent a promising approach for the controlled simultaneous delivery of CBD in combination with conventional chemotherapeutics. Several nanoformulations were designed and their effectiveness was proven in pre-clinical models, but this kind of study is still very scarce (Section 7, Table 3). Obviously, every new CBD formulation requires a range of pre-clinical studies in animals, which includes the evaluation of optimal administration routes, pharmacokinetics/pharmacodynamics, biodistribution, tissue and cancer cell specificity, stability, and safety. To confirm the anticancer efficacy of new formulations, cancer-specific pre-clinical models will be required, which may include chemically or genetically induced animal models, tumor allografts, and xenografts of human tumors in immunodeficient mice. Taking into consideration the high heterogeneity of cancer clones, experiments with patient-derived cancer tissue/cells are very desirable at this phase, to confirm the efficiency of CBD against specific cancer types. After satisfactorily completing all these pre-clinical studies, double-blind, randomized, placebo-controlled clinical studies could be performed. The combined use of CBD as both an antitumor and palliative agent is very attractive. Such an approach may be complicated by the fact that the effective concentrations, formulations, and administration routes are likely to be different for these two purposes. The observed synergism of CBD with conventional anticancer drugs can decrease the efficient drug and CBD concentrations, thus optimizing the treatment (Section 3). Importantly, CBD products’ quality should be controlled and self-medication should be inhibited, to prevent inappropriate use.

## Figures and Tables

**Figure 1 pharmaceuticals-15-00366-f001:**
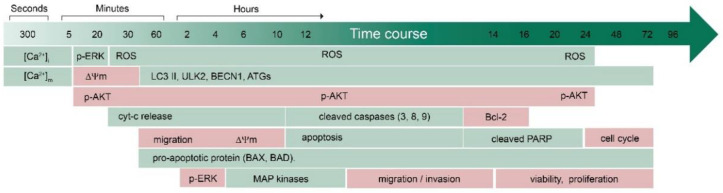
Time course of CBD-triggered changes in cellular processes in experimental cancer models in vitro. Increased activity is shown in green, whereas decreased activity is shown in pink.

**Figure 2 pharmaceuticals-15-00366-f002:**
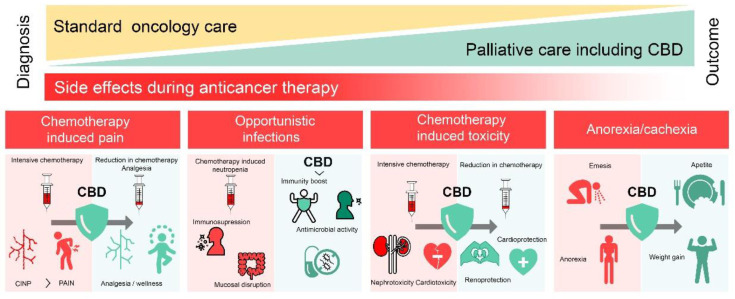
CBD can improve side effects of anticancer chemotherapy.

**Figure 3 pharmaceuticals-15-00366-f003:**
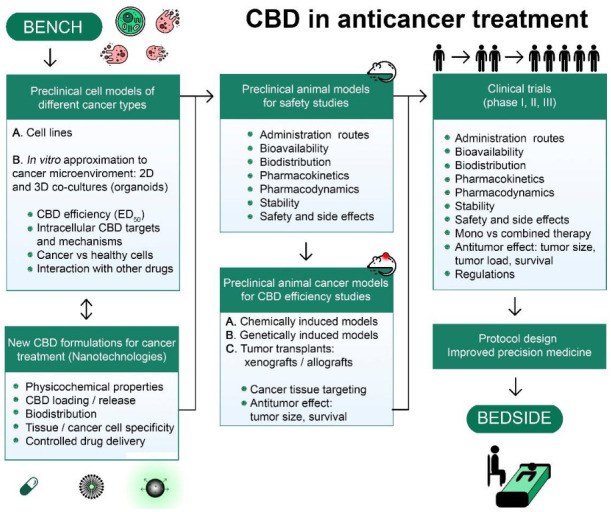
CBD path from the bench to the bedside.

**Table 1 pharmaceuticals-15-00366-t001:** Synergism of CBD with conventional chemotherapeuticals in cancer treatment.

Cancer Type	Experimental Model	Chemotherapeuticals Employed	Combination Index (CI)	Synergistic Effects	Proposed Mechanism	References
Bladder cancer	Cell lines: T24	Gemcitabine (0–20 μM) Cysplatin (0–100 μM) CBD (0–30 μM)	ND	↑ Cytotoxicity	ND	[62]
Breast cancer	Cell lines: MCF-7, MDA-MB-231	Doxorubicin (0–20 μM) Paclitaxel (0–500 nM) CBD (0–20 μM)	MDA-MB-231 0.59–0.83 MCF-7: 0.54–0.63	↑ Cytotoxicity	ND	[63]
Breast cancer	Xenograft: MDA-MB-231, 4T1	Paclitaxel (2.5–35 μM) CBD (2.7–4 μM)	MDA-MB-231: 4T1: 0.6–0.4 4T1: 0.9–0.8	↑ Cytotoxicity ↓ Tumor volume	5HT_1A_ receptors	[64]
Glioma	Xenograft: U87MG	Temozolomide (5 mg/kg) CBD (3.7 mg/kg)	0.78–0.887	↓ Tumor volume ↓ Tumor weight	Autophagy-mediated cell death	[11]
Glioma	Cell lines: U87MG, MZC	Doxorubicin (0–200 nM) Temozolomide (0–400 μM) Carmustine (0–200 μM) CBD (10 μM)	ND	↑ Cytotoxicity	TRPV2 overexpression TRPV2 activation	[65]
Glioma	Cell lines: T98G, U251, U87MG,	Temozolomide (1–1000 μM) Carmustine (3–1000 μM) Cisplatin (0–1000 μM) CBD (1–10 μM)	ND	↓ Proliferation ↑ Cytotoxicity	ND	[66]
Leukemia	Cell lines: CCFR-CEM, HL60	Cytarabine (5.4 μM) Vincristine (1.9 nM) CBD (4 μM)	CCFR:CEM: 0.92–0.61 HL60: 0.43-0.034	↑ Cytotoxicity	ND	[10]
Medulloblastoma Ependymoma	Cell lines: D283, D425, PER547	Cyclophosphamide (0–20 μM) CBD (0–7 μM)	ND	↑ Cytotoxicity ↑ Cell cycle arrest ↑ Apoptosis	ND	[45]
Multiple myeloma	Cell lines: U266, RPMI8226	Carfilzomib (0–100 nM) CBD (12.5 μM)	Specified as CI < 1	↑ Cytotoxicity	Apoptosis induction	[69]
	Synergism with other cytotoxic compounds					
Glioma	Cell lines: U87MG Xenograft: U87MG	THC (0–3.5 μM) CBD (0–3.5 μM)	ND	↑ Cytotoxicity ↓ Tumor volume ↓ Tumor weight	Autophagy-mediated cell death	[11]
Glioma	Cell lines: GSC387, GSC3832	Erastin (2.5–10 μM) Piperazine erastin (10 μM) CBD (0–10 μM)	GSC387: 0.64 GSC387: 0.53 GSC3832: 0.52	↑ ROS ↓ Tumor cell ↓ Invasion	ROS-mediated SLC7A11 upregulation	[68]
Glioma	Cell lines: U251, SF126	THC (0–5.4 μM) CBD (0–1.4 μM)	SF126: 0.22 U251: 0.29–0.27	↓ Cell growth ↑ Caspase activation ↑ Apoptosis	ERK inhibition	[15]
Glioma	Cell lines: U87MG, T98G Orthotopic tumor: GL261 in C57BL6 mice	CBD (0–20 μM) Irradiation (0–5 Gy)	U87MG: 0.9–08 T98G: 0.9–0.8 GL261: 0.9	↑ Cytotoxicity ↑ Autophagy ↓ Tumor volume ↓ Tumor progression	MAPK signaling	[70]
Leukemia	Cell lines: CCFR-CEM, HL60	THC (0–50 μM) CBD (0–50 μM)	CCRF-CEM: 0.53–0.44 HL60: 0.34–0.29	↑ Cytotoxicity	ND	[10]
Multiple myeloma	Cell lines: U266, RPMI8226	THC (12.5–50 μM) CBD (0–50 μM)	Specified as C < 1	↑ Cytotoxicity	Cell cycle arrest Autophagic cell death	[69]
Medulloblastoma Ependymoma	Cell lines: D283, PER547	THC (0–10.5 μM) CBD (0–9.5 μM)	ND	↑ Cytotoxicity ↑ Cell cycle arrest ↑ Autophagy	ROS-dependent mediated autophagy and apoptosis	[45]

**Table 3 pharmaceuticals-15-00366-t003:** Novel formulations proposed for cannabinoid delivery.

Carrier System	Structural Details	Models Tested	Administration Route	Advantages	Concerns and Limitations	Reference
Inorganic nanoparticles	Gold drones loaded with CBD	In vivo: transgenic mouse model bearing lung adenocarcinoma	Inhalation i.v.	Improved: Stability Bioavailability Retention in tumors	Loading concentration Drone size for EPR	[161]
Nano-micelles	Poly(styrene-co-maleic anhydride), cumene-terminated (SMA) micelles loaded with WIN	In vitro: breast cancer cell lines	Added to growth medium	Improved: Stability Bioavailability Retention in tumors	Loading concentration Micelle size for EPR	[162]
		In vivo: Female Balb/c mice bearing 4T1 mammary carcinoma	i.v.			
Polymeric microparticles	CBD-loaded poly-ε-caprolactone microparticles	In vivo: murine xenograft (glioblastoma) model	Local delivery	Long-lasting CBD delivery	Optimal particle size for better drug delivery	[163]
	CBD-loaded PLGA microparticles (25 μM)	In vitro and in ovo: breast or ovarian cancer cell lines	Added to growth medium or inoculated in chicken embryos	PLGA is FDA-approved Long-lasting delivery Possibility for multi-drug codelivery	Particle sterilization caused polymer erosion Particle size should be optimized to be suitable for bloodstream circulation	[164,165]
Lipid nanoparticles	CBD-loaded and CBD-decorated (functionalized) lipid nanoparticles	In vitro: glioma cell lines	Added to growth medium	Enhanced targeting and crossing of BBB Enhanced tumor targeting Biocompatible Biodegradable	Nanoparticle stability in organism	[166,167]
		In vivo: murine xenograft (glioma) model	i.v.			
Proteinoid nanoparticles	CBD-loaded Poly(RGD) proteinoid nanoparticles	In vitro: Colon carcinoma and breast cancer Cell lines	Added to growth medium	Cancer tissue targeting		[168]
		In vivo: Athymic mice bearing colon and breast cancer xenografts	i.v.			

## Data Availability

Not applicable.

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
