# Peer review of "Cannabidiol on the Path from the Lab to the Cancer Patient: Opportunities and Challenges"

_pharmaceuticals, 2022, doi:10.3390/ph15030366_

Round 1

Reviewer 1 Report

Excellent figure 1 "Time course of CBD-triggered changes in cellular processes in experimental cancer models in vitro" with increased activity in green, whereas decreased activity in pink.

The authors intelligently underline that clinical researchers, physicians and FDA expressed their concern that many patients use the variety of cannabis oils or whole plant extract of questionable quality (not of pharmaceutical grade) in self-prescribed dosages, what may be ineffective or even harmful for patients. Authors restrict them-selves to the most prominent and reliable data concerning toxicity and adverse effects of CBD.

They also discuss the importance of issues of use of medical CBD for cancer patients: 1) CBD formulations and administration ways to reach the desirable cytotoxic effect specifically in the cancer tissue, or favorable effects in palliative care; 2) possible side effects for specific CBD formulations and concentrations administrated by any specific route.

Table 1 in appendix A, is of particular interest as it lists the major paper with time course and studied target for each phenotype. One very recent paper ((Molecules 2022, 27(4), 1214; doi: 10.3390/molecules27041214) could be added to that highly informative listing, as this new paper (feb 22) concerns pancreatic human cancer a mighty resistant phenotype missing.

In # 8 (Regulation issues) the European countries should be added as CBD is legal in almost every European country since November 2020. Most European countries recognize the harmlessness of CBD and therefore allow CBD products containing it. It has been the case with Switzerland for nearly a decade and with Italy, Greece, Croatia, Spain, the Czech Republic, Estonia, and even Germany. More recently, end of December 2021, France has joined the list. (https://www.legalreader.com/cbd-in-europe-legal-status-of-cbd-country-by-country/)

Author Response

Many thanks for a rather positive and detailed evaluation.

Addresing your specific queries:

"Table 1 in appendix A, is of particular interest as it lists the major paper with time course and studied target for each phenotype. One very recent paper ((Molecules 2022, 27(4), 1214; doi: 10.3390/molecules27041214) could be added to that highly informative listing, as this new paper (feb 22) concerns pancreatic human cancer a mighty resistant phenotype missing."

Accepted, we have added the data from this paper to the Table A1, four entries at different timepoints.

In # 8 (Regulation issues) the European countries should be added as CBD is legal in almost every European country since November 2020. Most European countries recognize the harmlessness of CBD and therefore allow CBD products containing it. It has been the case with Switzerland for nearly a decade and with Italy, Greece, Croatia, Spain, the Czech Republic, Estonia, and even Germany. More recently, end of December 2021, France has joined the list. (https://www.legalreader.com/cbd-in-europe-legal-status-of-cbd-country-by-country/)

Many thanks for this link, it is very handy, indeed. We have added the sentence quoting the situation with CBD in Europe and respective link in chapter 8.

Reviewer 2 Report

This an interesting review articles on Cannabidiol and its therapeutic application. The manuscript is well written and conceptualized. The information curated and presented is very beneficial for the readers as it covers various topics around use of CBD in pre-clinical, clinical settings in different aspects of cancer and palliative care. However, there are some minor concerns which are as follows:

  1. English can be improved in the article.
  2. The research article can also be included where the role of CBD receptor TRPV2 and use of CBD is discussed in drug resistant cancers for example :-  doi.org/10.3390/cancers14051181,  doi: 10.3389/fphar.2021.746628,
  3. In Table 1 Combination Index values should be include if known.

Author Response

We are grateful for your helpful comments.

Herewith we are addressing specific points, raised by you:

  1. English can be improved in the article.

The text was double-checked for English style and grammar, necessary corrections have been made.

2. The research article can also be included where the role of CBD receptor TRPV2 and use of CBD is discussed in drug resistant cancers for example :-  doi.org/10.3390/cancers14051181,  doi: 10.3389/fphar.2021.746628,

Accepted. The first paper is now quoted, discussed in the text and respective data have been added to the Table A1. Second paper does not mention CBD at any place. But we have found an additional relevant paper on TRPV2 as a CBD target in triple negative breast cancer, which is now discussed (Chapter 2).

3. In Table 1 Combination Index values should be include if known.

Accepted. We have modified the Table 1 accordingly. We have also included the reference to the classical paper by Chou & Talalay (1984), which introduced the concept of CI, in case that some readers are unfamiliar with this index (Chapter 3).